# Endocarditis with *Streptococcus pseudoporcinus* Associated with Mastocytosis and Spondylodiscitis—A Coincidental Association? A Case Report

**DOI:** 10.3390/tropicalmed8050247

**Published:** 2023-04-25

**Authors:** Victoria Birlutiu, Rares-Mircea Birlutiu, Minodora Teodoru, Alina Camelia Catana, Cristian Ioan Stoica

**Affiliations:** 1Faculty of Medicine, Lucian Blaga University of Sibiu, County Clinical Emergency Hospital Sibiu, Str. Lucian Blaga, Nr. 2A, 550169 Sibiu, Romania; 2Clinical Hospital of Orthopedics, Traumatology, and Osteoarticular TB Bucharest, B-dul Ferdinand 35-37, Sector 2, 021382 Bucharest, Romania; 3County Clinical Emergency Hospital Sibiu, 550245 Sibiu, Romania; 4“Carol Davila” Faculty of Medicine, University of Medicine and Pharmacy, 050474 Bucharest, Romania

**Keywords:** infective endocarditis, *Streptococcus pseudoporcinus*, systemic mastocytosis, spondylodiscitis, case report

## Abstract

*Streptococcus pseudoporcinus* is a nonmotile Gram-positive, catalase, and benzidine negative, arranged in short chains, isolated from the genitourinary tract group B *Streptococcus*. *S. pseudoporcinus* was also identified from blood, urine, skin, cervical area, wounds, rectum, and placenta samples. Two cases of infective endocarditis have been reported in the literature. Based on these data, the identification of a case of *S. pseudoporcinus* infective endocarditis associated with spondylodiscitis in a patient with undiagnosed systemic mastocytosis until the age of 63 years is unusual. Two sets of blood specimens were collected, and both sets were positive for *S. pseudoporcinus*. Transesophageal echocardiography revealed, multiple vegetations on the mitral valve. A lumbar spine MRI revealed L5-S1 spondylodiscitis that associates prevertebral and right paramedian epidural abscesses with compressive stenosis. The performed bone marrow biopsy, and cellularity examination revealed 5–10% mast cells in the areas of medullary tissue, an aspect that is suggestive of mastocytosis. Antibiotic therapy was initiated, under which the patient presented intermittent fever. A second transesophageal echocardiography revealed a mitral valve abscess. A mitral valve replacement with a mechanical heart valve device through a minimally invasive approach was performed, with a favorable evolution under treatment. *S. pseudoporcinus* can be responsible for infectious endocarditis in certain immunodepressed cases, but also in a profibrotic, proatherogenic field, as shown by the association with mastocytosis in the presented case.

## 1. Introduction

*Streptococcus pseudoporcinus*, has been reported in human pathology since 2006 when it was differentiated from *Streptococcus porcinus*, being isolated from the female genitourinary tract. Few cases of infections have been published up to this point, either localized infection, a case of post-traumatic wound infection [1], a case of lower limbs cellulitis [2], or in the last years, cases of systemic infections that suggest a significant virulence of this species, that should not be neglected. Two cases of infective endocarditis have been reported in the literature [3,4], a case of pneumonia complicated with empyema associated with *Prevotella oris* [3], and a case of bacteremia in an immunosuppressed patient with syphilis and HIV co-infections [5]. Vergadi et al. report a difficult-to-treat case of cellulitis associated with bacteremia in a pediatric patient with Klippel-Trenaunay syndrome [6], Liatsos et al. report spontaneous peritonitis in a patient with liver cirrhosis with unfavorable outcome [7], and more recently, Dong et al. describe a case of orbital cellulitis resulting in corneal perforation, that required both antibiotic therapy and surgical intervention [8]. 

*S. pseudoporcinus*, a *β*-haemolytic *Streptococcus* of Lancefield groups E, P, V, U, NG1 (A1, C1), NG2, and NG3 [9], is a nonmotile Gram-positive, catalase, and benzidine negative, arranged in short chains, coccus, confused with group B *Streptococcus*, being isolated from the genitourinary tract like it (in Québec, Canada, 9 strains were differentiated between 1995–2005, from *S. porcinus*, by 16S rRNA gene sequencing). It has a spherical or ovoid shape and causes complete lysis on the Blood Agar culture medium, with an optimal growth between 10 °C and 45 °C (the temperature at which it no longer grows) being differentiated from *S. porcinus* by Bekal et al. in 2006 from a genetic point of view, using 16S rRNA gene sequencing [10]. Apart from its isolation from the genitourinary tract, *S. pseudoporcinus* was also identified from blood, urine, skin, cervical area, wounds, rectum, and placenta samples [9,10,11,12]. Genitourinary tract colonization appears to be more frequent in African American race, older age, and unmarried women, associated with poor education, local contraceptive measures (vaginal spermicides), and use of vaginal tampons. It is also reported the association with *Trichomonas vaginalis* infection or genital herpes [13]. Colonization with *S. pseudoporcinus* can be associated with smoking and a body mass index over 35 kg/m^2^, in pregnant women being responsible for premature or spontaneous premature birth [14]. Risk factors for a poor outcome are age, diabetes mellitus, high blood pressure, congestive heart failure, and immunosuppression [3,4,5,15,16]. In terms of antibiotic susceptibility testing, *S. pseudoporcinus* is susceptible to beta-lactams, macrolides, glycopeptides, sulfamethoxazole/trimethoprim, and clindamycin, and is resistant to tetracycline [11]. Liatsos et al. isolates a multidrug resistance strain, a strain that was resistant to penicillin, third-generation cephalosporins, and carbapenems [7].

Based on these data, the identification of a case of *S. pseudoporcinus* infective endocarditis associated with spondylodiscitis in a patient with undiagnosed systemic mastocytosis until the age of 63 years is unusual.

## 2. Case Report

We describe the case of a 63-year-old Caucasian male patient, who lives in North America and Europe, and seeks medical advice for prolonged fever (38–39.8 °C), weight loss of 10 kg in 2 months, marked physical asthenia, and pain on the mobilization of the lumbar spine. In his past medical history, the patient was a heavy smoker, with a 45 pack-year cigarette smoking history; he was also hypertensive, with functional capacity IV, and objective assessment C heart failure using the New York Heart Association (NYHA) classification system, right bundle branch block, and first-degree atrioventricular heart block. At the time of admission, on physical examination, the following changes were noticed: thoracoabdominal erythematous papular infiltrative rash, holosystolic murmur, hepatosplenomegaly, and hypogastric abdominal tenderness. In the context of a prolonged febrile illness, that was not investigated until that moment for any possible etiologies, an investigation protocol was initiated by a multidisciplinary team (infectious disease specialist, hematologist, surgeon, cardiologist) that included laboratory and complementary imaging investigations.

The main laboratory examinations that were performed are presented in Table 1 and are highlighting an important biological inflammatory syndrome. Two sets of blood specimens drawn 12 h apart were collected from independent venipuncture sites during the day of admission, and both sets were positive for *S. pseudoporcinus*, a strain that was sensitive to ampicillin, cefotaxime, ceftriaxone, clindamycin, chloramphenicol, erythromycin, tetracycline, vancomycin, moxifloxacin, levofloxacin, linezolid, tigecycline and intermediately sensitive to penicillin. An automated blood culture system that includes BACT/ALERT® 3D and BACT/ALERT® Culture Media (bioMérieux, Marcy-l’Étoile, France) was used. The isolated bacteria were identified using a VITEK 2 Compact analyzer (bioMérieux, Marcy-l’Étoile, France). Minimum inhibitory concentrations were assessed according to the EUCAST (European Committee on Antimicrobial Susceptibility Testing) breakpoints.

Transthoracic echocardiography was also performed and revealed the following data: intact interatrial septum, free left atrial appendage, mobile mitral valve—posterior leaflet shows prolapse and two mobile vegetations attached to the middle scallop (P2) segment. Regarding the first vegetation: oval-shaped vegetative formation of 15/8 mm with prolapse through the valve into the left ventricle. Regarding the second vegetation: filamentous mobile vegetative formation of 12 mm. Echogenic anterior mitral valve leaf with a 9.5 mm vegetation that was attached to the anteromiddle (A2) segment, without chordae breakage, severe mitral regurgitation jets. First vegetation prolapse and posterior mitral valve prolapse with 2 jets with vena contracta width of 5- and 3.5-mm. Conclusions: Multiple vegetations on the anterior and posterior leaflets of the mitral valve. Massive mitral insufficiency. Also, a transesophageal echocardiography was performed, and detailed information is reported in Figure 1.

ECG revealed frequent premature ventricular contractions (PVCs), (Figure 1A, black arrowheads) including couplets (Figure 1A, connected arrowheads), trigeminate PVCs (Figure 1B, black arrowheads), and an isolated premature supraventricular complex—PSVC (Figure 1B white line showing premature atrial depolarization, white arrowhead showing subsequent ventricular depolarization). Transthoracic echography revealed prolapse of the posterior mitral valve (VM), which presented two mobile vegetations attached to P2 and one mobile vegetation on the anterior mitral valve associated with severe mitral regurgitation. This was followed up by transesophageal echocardiography, which better defined the two vegetations on the posterior mitral valve: one measuring approximately 12 mm in diameter (Figure 1C—ME 2 chambers) and the other measuring approximately 15/8 mm (Figure 1D—ME 2 chambers) and prolapsing through the mitral orifice; as well as the vegetation on the anterior mitral valve (9.5 mm diameter) (Figure 1E—ME LAX). In addition, the posterior mitral valve showed prolapse. Mitral regurgitation was quantified as severe, with two jets (Figure 1F—ME 2 chambers, color doppler; green arrowheads show the two jets, VC1 = 5 mm; VC2 = 3.5 mm). Transthoracic echocardiographic reevaluation one week later revealed a new vegetation at the base of the posterior mitral valve with dimensions of approximately 10/11 mm (Figure 1G—A4C) accompanied by a new mitral regurgitation jet at the same location (Figure 1H—A4C, color doppler; green arrowhead showing previous central jet, green arrow showing newly discovered jet, vena contracta = 3 mm), raising the suspicion of a valvular abscess with consequent perforation. Surgery for mitral mechanical valve implantation occurred without complications (Figure 1I showing the postoperative transthoracic echocardiographic aspect of the mitral valve; A4C with continuous doppler flow measuring mitral valve velocity time integral (MV-VTI)).

A lumbar spine magnetic resonance imaging (MRI) was also performed, and the following results were reported: narrowing of the L5-S1 intervertebral space with discretely irregular edges of the adjacent vertebral end plates, with moderate disc edema and "mirror" edema at the level of the vertebral end plates. Linear progressive contrast uptake at the level of the vertebral end plateaus adjacent to the L5-S1 intervertebral disc, with fluid collections with contrast outline uptake at the anterior prevertebral L5-S1 soft tissues. At a posterior paramedian right intervertebral disc and epidural level compressive stenosis. Conclusions: patchy gadolinium uptake edema, L5-S1 spondylodiscitis that associates prevertebral and right paramedian epidural abscesses with compressive stenosis. A T2-weighted image from the MRI study is reported in Figure 2.

Laboratory examinations also reveal hypergammaglobulinemia (increased IgG and IgA levels), and nonmonoclonal increased kappa and lambda light chains. Serum tryptase levels of 69μg/L (reference values 0–20μg/L), and peripheral blood c-KIT were slightly positive. A bone marrow biopsy was performed, and cellularity examination revealed 5–10% mast cells in the areas of medullary tissue, an aspect that is suggestive of mastocytosis. It was recommended to assess the result together with the anatomopathological examination and immunophenotyping for the most accurate assessment of mastocytosis. Immunophenotyping was performed using the CD45, CD117, CD34, CD203, HLADR, CD123 panel that identified the following cell populations: Mast cells CD117+/CD203+, CD34−/HLADR−, CD123−, CD25+, CD2+ 0.15% mast cells. The conclusion of the immunophenotyping examination was that the bone marrow is infiltrated with mature, pathological CD25+/CD2+ mast cells, in a percentage of 0.15%. Bone marrow biopsy aspects are reported in Figure 3.

Bone marrow biopsy anatomopathological examination reveals a focal reactive bone marrow type I aspect with mast cell infiltrates. Skin biopsy does not reveal mast cell proliferation on the examined specimens. Detection of the c-KIT mutation was performed by molecular biology and was weakly positive for D816V.

From sputum, sperm, and urine culture *Escherichia Coli* was isolated, and also *Enterobacter aerogenes* was isolated from sputum cultures. Both isolated strains were susceptible to 3rd- and 4th-generation cephalosporins, carbapenems, aminoglycosides, and piperacillin-tazobactam. Antibiotic therapy was initiated with ceftriaxone 2 g/day and vancomycin 2 × 1 g/day, under which the patient presented intermittent fever. 10 days after the first transethoracic echocardiography, a second one is performed and revealed a hyperechoic, oval-shaped formation of 11/10 mm at the base of the posterior leaflet of the mitral valve, an aspect that might suggest an abscess, with a possible valve perforation at this level, mitral regurgitation through 3 jets: central, eccentric and a jet through the back leaflet (perforation?) with vena contracta width of 3 mm. 

The decision to undergo surgical intervention was taken, and the patient was referred to the cardiovascular surgical department, where a mitral valve replacement with a St. Jude Medical no. 31 mechanical heart valve fixed with COR-KNOT device through a minimally invasive approach was performed. From the harvested hemocultures prior to the surgery and the bacteriological examination of the vegetation no pathogens were isolated. Histopathological examination of the resected tissue revealed: a white membraniform tissue fragment, with an irregular contour, elastic consistency, and 3 × 2 × 1 cm in size. On microscopic examination, the examined tissue is represented by a fibro-conjunctival structure with collagen fibers, with hyalinization and focal calcifications, that associates acute and chronic non-specific adjacent inflammatory process, with the giant-cellular granulomatous process, rich vascularization and fibrosis (reparative/scarring context). The treatment with ceftriaxone was continued for up to 4 weeks and later, at discharge, with levofloxacin 750 mg/day, another 2 months for the L5-S1 spondylodiscitis, with a favorable evolution under treatment. The patient remained under cardiological follow-up and is under long-term therapy with anticoagulant, antiplatelet, hypotensive, and statin therapy. The favorable evolution of the patient was the result of an interdisciplinary collaboration between multiple specialists, who helped caring this case.

## 3. Discussion

Mastocytosis is a condition associated with the proliferation and excessive accumulation of cutaneous (cutaneous mastocytosis) or systemic mastocytes, which is responsible for the activation and release of vasoactive cellular mediators (histamine, cytokine, tumor necrosis factor, growth factors, protease, and phospholipases), triggering a polymorphism of symptoms from anaphylaxis, to pruritus, or gastrointestinal manifestations (nausea, vomiting, diarrhea, abdominal pain, peptic ulcer, gastrointestinal bleeding, etc.), musculoskeletal disorders (bone and muscle pain similar to fibromyalgia), osteopenia/osteoporosis, psychiatric manifestations (the so-called “mixed organic brain syndrome”), lymphadenopathy, hepatosplenomegaly, hematological changes (anemia, and less often eosinophilia). The real incidence of mastocytosis is not known, but it affects both genders equally, the cutaneous form is mostly diagnosed in childhood, while the systemic form is present in 95% of cases in adults [17]. The most common genetic mutation responsible for systemic mastocytosis is KIT D816V, less often other mutations such as FIP1L1-PDGFR, TET2, IgE, JAK2 V617F, and RAS are also responsible. Cardiac involvement may be suggested by recurrent syncope, Kounis syndrome (acute coronary syndrome induced by mast cell degranulation by coronary spasm, coronary artery occlusion, or rupture of preexisting atherosclerotic plaque secondary to mast cell activation) [18,19], increased risk of myocardial infarction [20], or sudden death [21]. The proatherogenic role of mastocytosis is recognized, and also an inducer of fibrogenesis [22,23].

The updated diagnostic criteria of mastocytosis are reported by Valent P et al. Major criterion: Multifocal dense infiltrates of mast cells (≥15 mast cells in aggregates) in bone marrow biopsies and/or in sections of other extracutaneous organ(s). Minor criteria: a. ≥25% of all mast cells are atypical cells (type I or type II) on bone marrow smears or are spindle-shaped in mast cell infiltrates detected in sections of bone marrow or other extracutanous organs; b. KIT-activating KIT point mutation(s) at codon 816 or in other critical regions of KITb in bone marrow or another extracutaneous organ; c. Mast cells in bone marrow, blood, or another extracutaneous organ express one or more of: CD2 and/or CD25 and/or CD30c; d. Baseline serum tryptase concentration >20 ng/mL (in the case of an unrelated myeloid neoplasm, an elevated tryptase does not count as an systemic mastocytosis criterion. In the case of a known hereditary alpha-tryptasemia, the tryptase level should be adjusted. If at least 1 major and 1 minor or 3 minor criteria are fulfilled → the diagnosis is systemic mastocytosis [24]

The diagnosis of systemic mastocytosis in our case report was established based on the presence of 1 major (the presence of mast cells in the bone marrow biopsy tissue) and 3 minor criteria (KIT-activating KIT point mutation at codon 816, CD25+ mast cells in bone marrow, and baseline serum tryptase concentration >20 ng/mL). Concomitant vertebral and cardiac impairment may be the consequence of the lesions associated with mastocytosis, which would explain the rapid evolution towards perforation of the mitral valve through the profibrotic pathways. Although it is the only case of systemic mastocytosis reported in the literature to date, which associates infective endocarditis caused by a pathogen rarely encountered in human pathology, we consider that it is important to include in the possible cardiovascular diseases associated with mastocytosis also infective endocarditis. The association of infective endocarditis with *S. pseudoporcinus* with pneumonia or pulmonary empyema [3], on aortic valve [4], associated with chronic obstructive pulmonary disease (COPD), mitral valve prolapses, and history of pulmonary embolism [25], in pregnant women [26] was described, but never in the infective endocarditis-spondylodiscitis-mastocytosis triad. In terms of the limitations of our case report, they are related to the lack of etiological diagnosis of spondylodiscitis, the case being treated conservatively and not surgically.

## 4. Conclusions

*S. pseudoporcinus* can be responsible for infectious endocarditis in certain immunodepressed cases, but also in a profibrotic, proatherogenic field, as shown by the association with mastocytosis in the presented case.

## Figures and Tables

**Figure 1 tropicalmed-08-00247-f001:**
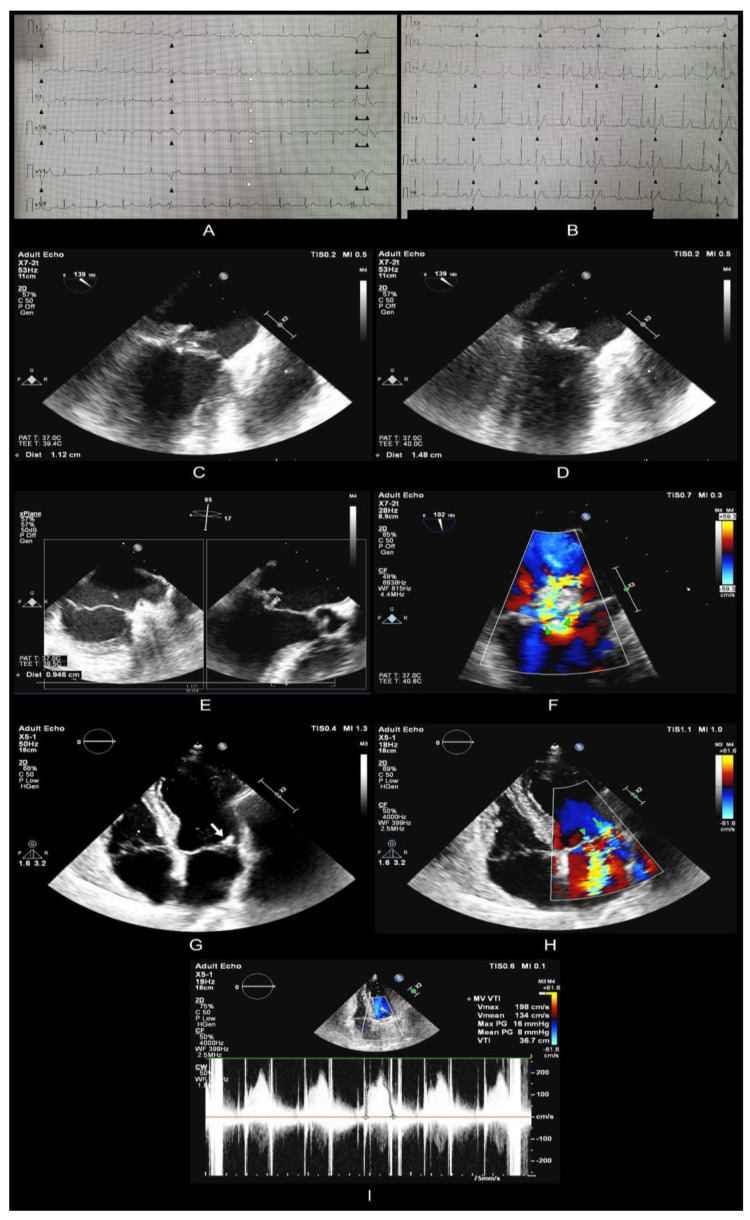
ECG, Transthoracic and transesophageal echocardiography examinations preoperative and postoperative.

**Figure 2 tropicalmed-08-00247-f002:**
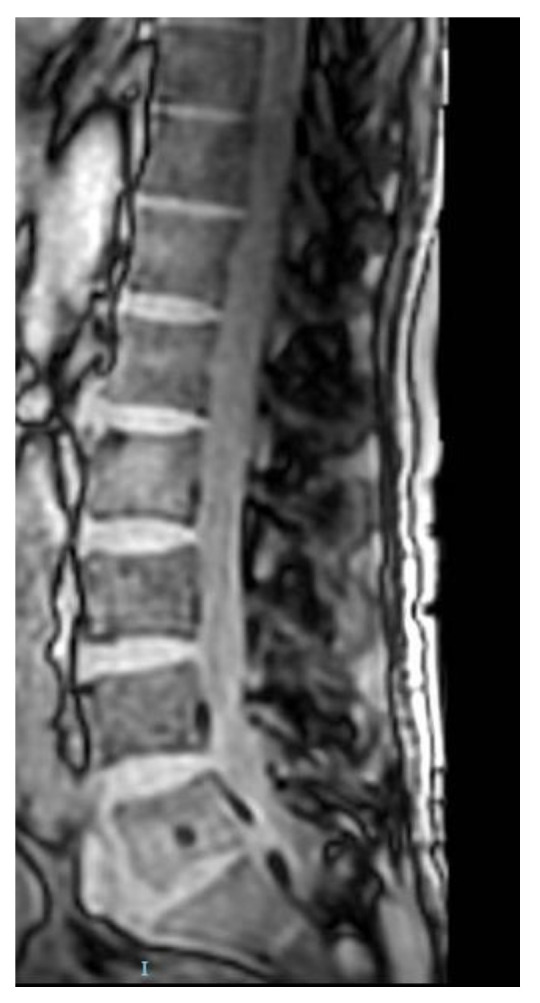
MRI scans of the lumbar spine. A T2-weighted image MRI studies.

**Figure 3 tropicalmed-08-00247-f003:**
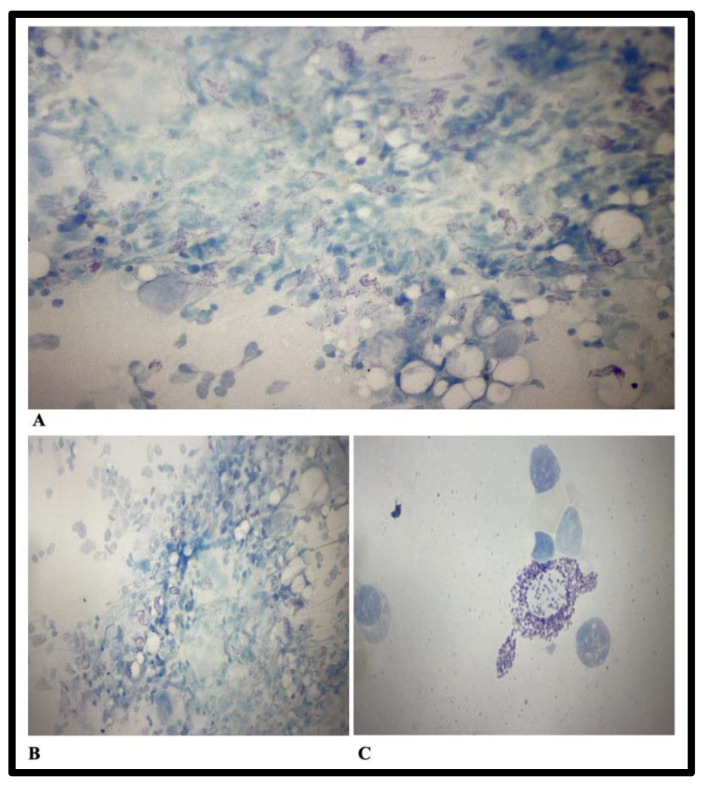
Bone marrow biopsy aspects and cellularity examination. (**A**,**B**)—May Grunwald-Giemsa, active macrophages in the medullary tissue areas, mast cells between 5–10%. (**C**)—May Grunwald-Giemsa staining.

**Table 1 tropicalmed-08-00247-t001:** Laboratory examinations during hospitalization.

Date	Parameter	Values	Reference Value
On admission	C-Reactive Protein	114.81 mg/L	0–5 mg/L
	Fibrinogen	514 mg/dL	170–420 mg/dL
	WBCs	14.70 × 103/µL	4.2–7.5 × 103/µL
	Differential blood count:		
	Neutrophils	11.77 × 103/µL	10 × 103/µL
	Lymphocytes	1.91 × 103/µL	1.5–4 × 03/µL
	Monocytes	0.95 × 103/µL	0.2–1 × 103/µL
	Basophils	0.02 × 103/µL	0–0.2 × 103/µL
	Eosinophils	0.05 × 103/µL	0–0.7 × 103/µL
	Red Blood Cells	3.35 × 10^6^/µL	4.5–5.8 × 10^6^/ µL
	Haemoglobin	10.3 g/dL	13–17 g/dL
	Hematocrit	30.6%	40–50%
	Thrombocytes	170 × 10^3^/uL	150–400 × 10^3^/uL
	Coagulation tests	Prothrombin time (PT):	
		13.9	9.9–12.3 s
		Activated partial thromboplastin time (aPTT):	
		30.9	25.1–37.7 s
		International normalized ratio (INR):	
		1.10	0.86–1.1
	Serum protein electrophoresis:		
	Albumin	50.1%	54.3–65.5%
	Alpha-1 globulins	4.1%	1.2–3.3%
	Alpha-2 globulins	9.1%	8.3–15%
	Beta- globulins	13.7%	8.6–14.8%
	Gamma- globulins	23%	7.1–19.5%
	IgA	432 mg/dL	70–400 mg/dL
	IgG	1661 mg/dL	700–1600 mg/dL
	IgM	133 mg/dL	40–230 mg/dL
	ESR*	74 mm/h	0–15 mm/h
	Urea	34 mg/dL	18–55 mg/dL
	Creatinine	0.95 mg/dL	0.72–1.25 mg/dL
	Blood glucose	128 mg/dL	80–115 mg/dL
	Urinalysis	Color —Yellow Clarity/turbidity—Clear pH—5.5 Specific gravity—1.015 Glucose—125 mg/d Ketones—None Nitrites—Negative Bilirubin—Negative Urobilirubin—Normal Blood—Negative Protein—100 mg/d RBCs—1–2 RBCs/hpf WBCs—2–3 WBCs/hpf Squamous epithelial cells—5–10 squamous epithelial cells/hpf	
	Urine Culture	*E. coli* >100,000 CFU/ML	
	HIV-1/HIV-2 Antibody Test	Negative	
	Rapid plasma reagin	Negative	
12 days after admission	C-Reactive Protein	26.77 mg/L	
	Fibrinogen	364 mg/dL	
	WBCs	18.69 × 103/µL	
	Haemoglobin	11.4 g/dL	
	Hematocrit	34.4 %	
	Thrombocytes	155 × 103/µL	
19 days after admission	C-Reactive Protein	26.45 mg/L	
	Fibrinogen	443.9 mg/dL	
	WBCs	14.93 × 103/µL	
	Haemoglobin	11.8 g/dL	
	Hematocrit	35.8 %	
	Thrombocytes	187 × 103/µL	
	ESR *	16 mm/h	

* ESR (erythrocyte sedimentation rate).

## Data Availability

All data generated or analyzed during this study are included in this published article.

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
