# Peer review of "Endocarditis with Streptococcus pseudoporcinus Associated with Mastocytosis and Spondylodiscitis—A Coincidental Association? A Case Report"

_tropicalmed, 2023, doi:10.3390/tropicalmed8050247_

Round 1

Reviewer 1 Report

77- degree and duration of fever? Is patient obese or smoker? 

83- how papular rash was interpreted?

92 - blood culture method, strain identification method

94 - antibiogram method

Table 1: RBC, urine culture, urinalysis, renal function, HIV, blood sugar, coagulation tests

109- ejection fraction?

139 - Why sputum, urine and semen culture were performed? Was a chest X-ray performed? How you interpret the results of these cultures? Was patient evaluated for a STD?

142 - lack of diagnosis criteria for endocarditis (modified Duke criteria)

142- Why ceftriaxone with vancomycin for treatment? (Endocarditis guidelines)

150 - culture from valve was performed during replacement?

153 - why levofloxacin for spondylodiscitis, why 750mg, why 2 months?

154 - Was a colonoscopy performed?

159 Discussions are mainly about mastocytosis and not about special etiology of endocarditis. Did not discuss about previous published cases of S. pseudoporcinus endocarditis/infections

Author Response

Sibiu, 2.04.2023

To

the Editors of Microorganisms®

Dear Editor,

Dear Reviewer,

Thank you for reviewing our manuscript. Please find attached a revised version of our manuscript, “Endocarditis with Streptococcus pseudoporcinus associated with mastocytosis and spondylodiscitis – a coincidental association? A case report?”.

Your and the reviewers’ comments were highly insightful and enabled us to greatly improve the quality of our manuscript. We have modified the manuscript in response to the comments. Attached is our point-by-point response to each comment.

Reviewer Comments:

Reviewer 1

Answer: Thank you for taking your precious time to be able to assess our manuscript. The comments were highly insightful and enabled us to improve our manuscript.

77- degree and duration of fever? Is patient obese or smoker?

Answer: The following information was added to the manuscript – (38-39.8°C), smoker with 45 pack-years. The patient had a normal BMI.

83- how papular rash was interpreted?

Answer: The conclusion of an examination carried out by a dermatologist is thoracoabdominal erythematous papular infiltrative rash, and recommended TPHA, tryptase, c-kit, Angiotensin converting enzyme;

92 - blood culture method, strain identification method

94 - antibiogram method

Answer: The following statement has been added to the manuscript “An automated blood culture system that includes BACT/ALERTâ 3D and BACT/ALERTâ Culture Media (bioMérieux, Marcy-l’Étoile, France) was used. The isolated bacteria were identified using a VITEK 2 Compact analyzer (bioMérieux, Marcy-l’Étoile, France). Minimum inhibitory concentrations were assessed according to the EUCAST (European Committee on Antimicrobial Susceptibility Testing) breakpoints.”

Table 1: RBC, urine culture, urinalysis, renal function, HIV, blood sugar, coagulation tests

Answer: The suggested laboratory data was added to Table no.1

109- ejection fraction?

Answer: 50%

139 - Why sputum, urine and semen culture were performed? Was a chest X-ray performed? How you interpret the results of these cultures? Was patient evaluated for a STD?

Answer: The patient was fully examined, which required medical advice for a prolonged febrile illness, and since during the clinical examination the patient reported also low back/ lumbar pain, hypogastrium pain, and disorders of micturition. The sputum was described by the patient as muco-purulent. A chest x-ray was also performed and revealed the following: Free lateral costodiaphragmatic recess, prominent pulmonary hilum, vascular type ones. Without radiographically detectable lung condensation processes. Discreet bilaterally accentuated pulmonary interstitium. A lower left heart contour, and tortuous aorta. Chronic degenerative spondylarthrosis changes in the dorsal spine. In the conditions of multiple overlaps, apparently without osteolytic lesions in the chest. Microbiological results of the cultures have been added to the manuscript. The RPR test was negative. HIV1/2 negative

142 - lack of diagnosis criteria for endocarditis (modified Duke criteria)

142- Why ceftriaxone with vancomycin for treatment? (Endocarditis guidelines)

Answer: The diagnosis of infective endocarditis was made according to the modified Duke criteria 2020, based on the major criteria, namely two positive blood cultures for organisms typical of endocarditis drawn 12 hours apart as we mentioned in the text, echocardiographic evidence of endocardial involvement - the presence of endocardial vegetation, cardiac abscess. The treatment with ceftriaxone and vancomycin was chosen for the penetrability of these antibiotics at the vertebral level - spondylodiscitis, and the absence of an etiology from this level (a biopsy and a bacteriological examination of the vertebral abscess was not performed).

150 - culture from valve was performed during replacement?

Answer: The patient underwent surgical intervention in a private service, we have no information regarding the bacteriological examination of the vegetation

153 - why levofloxacin for spondylodiscitis, why 750mg, why 2 months?

Answer: In the absence of a bacteriological examination from the vertebral abscess, we could not support the same streptococcal etiology, which is why the antibiotic therapy was continued with levofloxacin for the antistaphylococcal activity, this being the most common etiology. The patient had an already scheduled travel to the US and required oral therapy. The duration of the therapy is 6-8 weeks. The same doses are found in the IDSA guidelines Elie F. Berbari, Souha S. Kanj, Todd J. Kowalski, Rabih O. Darouiche, Andreas F. Widmer, Steven K. Schmitt, Edward F. Hendershot, Paul D. Holtom, Paul M. Huddleston, III, Gregory W. Petermann, Douglas R. Osmon, 2015 Infectious Diseases Society of America (IDSA) Clinical Practice Guidelines for the Diagnosis and Treatment of Native Vertebral Osteomyelitis in Adults, Clinical Infectious Diseases, Volume 61, Issue 6, 15 September 2015, Pages e26–e46, https://doi.org/10.1093/cid/civ482

154 - Was a colonoscopy performed?

Answer: A colonoscopy was not performed. We did not consider it necessary. The patient was on anticoagulant therapy, etc.

159 Discussions are mainly about mastocytosis and not about special etiology of endocarditis. Did not discuss about previous published cases of S. pseudoporcinus endocarditis/infections

Answer: Thank you for the question. We mentioned the previous 2 published cases and reported what was particular. Infective endocarditis is present in 3-10 cases/100.000 people (Cahill TJ, Prendergast BD. Infective endocarditis. Lancet. 2016 Feb 27;387(10021):882-93). Etiology of infective endocarditis (AW Yallowitz , LC Decker- Infectious Endocarditis NCBI Bookshelf) is dominated by streptococci (especially from the oral flora), staphylococci, and enterococci, followed by the HACEK group (Haemophilus, Actinobacillus, Cardiobacterium, Eikenella, and Kingella), only in 6% of cases other bacteria are identified (Staph. epidermidis associated is with cardiovascular devices or IV drug administration, S. gallolyticus in patients with colon carcinoma especially, Coxiella burnetti, etc.)

We hope that the revised form of the manuscript and our accompanying responses will be sufficient to make our manuscript suitable and accepted for publication in Microorganisms®. We shall look forward to hearing from you at your earliest convenience.

With our best regards,

Sincerely yours,

Victoria Birlutiu, Prof. Habil. MD. PhD

Rares Mircea Birlutiu, MD PhD

Reviewer 2 Report

This is a very interesting case report where the authors showed the case of endocarditis with Streptococcus pseudoporcinus associated with systemic mastocytosis and spondylodiscitis. However, as a single case report, the manuscript should include more clinical data for future references. Also, the authors should rewrite some of the sections for more clarity and longer sentences.

1. In the abstract it is mentioned that there are two cases of infective endocarditis. However, in the manuscript only one case has been reported. 

2. The results section is descriptive and can be concise with the addition of tables and figures. For example, figures can be shown for echocardiography and MRI results. Also, a table can be generated from the microbiological report from sputum, sperm, urine and blood samples.

3. The authors can discuss their finding in relation to the previous clinical reports. 

Author Response

Sibiu, 2.04.2023

To

the Editors of Microorganisms®

Dear Editor,

Dear Reviewer,

Thank you for reviewing our manuscript. Please find attached a revised version of our manuscript, “Endocarditis with Streptococcus pseudoporcinus associated with mastocytosis and spondylodiscitis – a coincidental association? A case report?”.

Your and the reviewers’ comments were highly insightful and enabled us to greatly improve the quality of our manuscript. We have modified the manuscript in response to the comments. Attached is our point-by-point response to each comment.

Reviewer Comments:

Reviewer 2

This is a very interesting case report where the authors showed the case of endocarditis with Streptococcus pseudoporcinus associated with systemic mastocytosis and spondylodiscitis. However, as a single case report, the manuscript should include more clinical data for future references. Also, the authors should rewrite some of the sections for more clarity and longer sentences.

Answer: Thank you for taking your precious time to be able to assess our manuscript. The comments were highly insightful and enabled us to improve our manuscript. We added more clinical data during this revision, also we were able to acquire some images from the hospital archive from the TTE, TEE, ECG, and MRI examinations, Also the SM criteria were included in the manuscript.

In the abstract it is mentioned that there are two cases of infective endocarditis. However, in the manuscript only one case has been reported.

Answer: Thank you for the question! We apologize if the statements from the manuscript are not clear regarding the two previously reported cases. We mentioned in the discussions section of the manuscript the main findings in two 2 case reports. Both cases are also mentioned as references (ref.3 and 4.) taking your precious time to be able to assess our manuscript. The comments were highly insightful and enabled us to improve our manuscript.

  1. The results section is descriptive and can be concise with the addition of tables and figures. For example, figures can be shown for echocardiography and MRI results. Also, a table can be generated from the microbiological report from sputum, sperm, urine and blood samples.

Answer: Part of the question was addressed in the first question answer. We included a table with the microbiological examination.

  1. The authors can discuss their finding in relation to the previous clinical reports.

Answer: The question was addressed in the first question answer.

We hope that the revised form of the manuscript and our accompanying responses will be sufficient to make our manuscript suitable and accepted for publication in Microorganisms®. We shall look forward to hearing from you at your earliest convenience.

With our best regards,

Sincerely yours,

Victoria Birlutiu, Prof. Habil. MD. PhD

Rares Mircea Birlutiu, MD PhD

Reviewer 3 Report

The authors report a rare case combining infective endocarditis due to Streptococcus pseudoporcinus to a profibrotic state such as mastocytosis. The case is well documented, except about imaging : transoesophagal echography (TEE) and magnetic resonance imaging (MRI).

Major comments :

1) Please add imaging documentation : TEE and MRI.

2) Please add a state of art about mastocytosis diagnostic criteria.

3) Redaction :

General : 

S. pseudoporcinus instead of Str. Pseudoporcinus

S. porcinus instead of Str. porcinus

Abstract :

Lines 28-29 : confuse…

Introduction :

Line 66 : diabetes mellitus instead of mellites.

Table 1 : imprecise redaction

Do not use ^ for exposants

µ instead of u

About numbers, choose . or ,

Not alfa, gama : alpha, gamma

Repeat units.

Precise date format.

Case report :

Bad redaction of TEE and MRI conclusions : too telephonic style.

Discussion : 

Line 185 : reported instead of published.

Define COPD.

Author Response

Sibiu, 2.04.2023

To

the Editors of Microorganisms®

Dear Editor,

Dear Reviewer,

Thank you for reviewing our manuscript. Please find attached a revised version of our manuscript, “Endocarditis with Streptococcus pseudoporcinus associated with mastocytosis and spondylodiscitis – a coincidental association? A case report?”.

Your and the reviewers’ comments were highly insightful and enabled us to greatly improve the quality of our manuscript. We have modified the manuscript in response to the comments. Attached is our point-by-point response to each comment.

Reviewer Comments:

Reviewer 3

The authors report a rare case combining infective endocarditis due to Streptococcus pseudoporcinus to a profibrotic state such as mastocytosis. The case is well documented, except about imaging : transoesophagal echography (TEE) and magnetic resonance imaging (MRI).

Answer: Thank you for taking your precious time to be able to assess our manuscript. The comments were highly insightful and enabled us to improve our manuscript.

Major comments :

1) Please add imaging documentation: TEE and MRI.

Answer: Thank you for the suggestion. We were able to add to the manuscript some images from the echography and also from the MRI examination of the spine.

2) Please add a state of art about mastocytosis diagnostic criteria.

Answer: A state of art about mastocytosis diagnostic criteria: As it is reported by Valent P, Akin C, Hartmann K, Alvarez-Twose I, Brockow K, Hermine O, Niedoszytko M, Schwaab J, Lyons JJ, Carter MC, Elberink HO, Butterfield JH, George TI, Greiner G, Ustun C, Bonadonna P, Sotlar K, Nilsson G, Jawhar M, Siebenhaar F, Broesby-Olsen S, Yavuz S, Zanotti R, Lange M, Nedoszytko B, Hoermann G, Castells M, Radia DH, Muñoz-Gonzalez JI, Sperr WR, Triggiani M, Kluin-Nelemans HC, Galli SJ, Schwartz LB, Reiter A, Orfao A, Gotlib J, Arock M, Horny HP, Metcalfe DD. Updated Diagnostic Criteria and Classification of Mast Cell Disorders: A Consensus Proposal. Hemasphere. 2021 Oct 13;5(11):e646. doi: 10.1097/HS9.0000000000000646, a major criteria that is represented by multifocal dense infiltrates of mast cells (≥15 mast cells in aggregates) in bone marrow biopsies and/or in sections of other extracutaneous organ(s) to which a minor criteria is required or 3 minor criteria like  ≥25% of all mast cells are atypical cells (type I or type II) on bone marrow smears or are spindle-shaped in mast cell infiltrates detected in sections of bone marrow or other extracutanous organs; KIT-activating KIT point mutation(s) at codon 816 or in other critical regions of KITb in bone marrow or another extracutaneous organ; Mast cells in bone marrow, blood, or another extracutaneous organ express one or more of: CD2 and/or CD25 and/or CD30; Baseline serum tryptase concentration >20 ng/mL (in the case of an unrelated myeloid neoplasm, an elevated tryptase does not count as an SM criterion are required. In the case of a known HαT, the tryptase level should be adjusted.

We also reported the criteria in the manuscript

3) Redaction :

General :

  1. pseudoporcinus instead of Str. Pseudoporcinus
  2. porcinus instead of Str. porcinus

Answer: Done! Thank you for pointing this fact!

Abstract : Lines 28-29 : confuse… Introduction : Line 66 : diabetes mellitus instead of mellites. Table 1 : imprecise redaction Do not use ^ for exposants; µ instead of u About numbers, choose . or , Not alfa, gama : alpha, gamma Repeat units. Precise date format. Case report: Bad redaction of TEE and MRI conclusions: too telephonic style. Discussion: Line 185: reported instead of published. Define COPD.

Answer: Done! Thank you for pointing this facts and for the suggestions! We hope that we addressed this issues!

We hope that the revised form of the manuscript and our accompanying responses will be sufficient to make our manuscript suitable and accepted for publication in Microorganisms®. We shall look forward to hearing from you at your earliest convenience.

With our best regards,

Sincerely yours,

Victoria Birlutiu, Prof. Habil. MD. PhD

Rares Mircea Birlutiu, MD PhD

Round 2

Reviewer 1 Report

Minor revision

Author Response

Sibiu, 5.04.2023

To

the Editors of Tropical Medicine and Infectious Disease®

Dear Editor,

Dear Reviewer,

Thank you for reviewing our manuscript. Please find attached a revised version of our manuscript, “Endocarditis with Streptococcus pseudoporcinus associated with mastocytosis and spondylodiscitis – a coincidental association? A case report?”.

Your and the reviewers’ comments on our revised manuscript were highly insightful and enabled us to greatly improve the quality of our manuscript. We have modified the manuscript in response to the comments. Attached is our point-by-point response to each comment.

Reviewer Comments:

Reviewer 1

Answer: Thank you for taking your precious time to be able to reassess our manuscript. The comments were highly insightful and enabled us to improve our manuscript.

smoker with 45 pack-years

Answer: was changed to “with a 45 pack-year cigarette smoking history”

Consider as colonization. No respiratory symptoms and no leukocytes in urine.

Answer: It might be, we can talk about biofilm-related infections in the case of colonization or not, but unfortunately it was not in this case, but might be in the future. Both Escherichia Coli and Enterobacter aerogenes can also be the pathogens involved in spondylodiscitis. In the setting of a prolonged febrile syndrome with systemic infection criteria, it is not usual not to take into consideration all the bacteriological results in the context of clinical manifestations.

Repeated information

Answer: The information is recurring because reviewer no.3 recommended inserting into the manuscript a table (table no.2) to highlight the isolated strains and the AST result. We try to comply with all reviewers.

PVCs

Answer: PVCs - Premature Ventricular Contractions

PSVC

Answer: PSVC - premature supraventricular complex

Comment about the coexisting infections: spondylodiscitis and infective endocarditis. The probability to have the same etiology is high.

Answer: In the absence of spinal surgery or at least a biopsy, the etiology of the infection remains unknown, clearly there may be two coincident pathologies or it may be a pathology caused by the same etiological agent.

All suggested format changes were performed.

We hope that the revised form of the manuscript and our accompanying responses will be sufficient to make our manuscript suitable and accepted for publication in Tropical Medicine and Infectious Disease®. We shall look forward to hearing from you at your earliest convenience.

With our best regards,

Sincerely yours,

Victoria Birlutiu, Prof. Habil. MD. PhD

Rares Mircea Birlutiu, MD PhD

Reviewer 2 Report

The revised version of the manuscript has improved a lot with the addition of new figures, and tables. The authors should work on a few minor things before the final publication.

Line 19: Use an abbreviated name for the organism (S. pseudoporcinus) as the full organism name appeared before. Similar in Lines 21, 29, 73, 94, 161, 252.

Line 38: S. porcinus appears first here. Use the full name.

Line 119: Add MRI in the parenthesis after magnetic resonance imaging. 

Line 161-163: This is the exact repeat of Lines 94-97.

Line 178: EEG or ECG?

ECG image figure number should be changed to Figure no. 3.

According to the organization of the manuscript, ECG figure (Figiure 3) should come first, followed by MRI (Figure 1) and biopsy images (Figure 2). 

Figure legends should go after the figures.

Italicize the scientific name of the organism in the reference.

Figures 1, 2 and Table 2 are not referred to in the texts.

Author Response

Sibiu, 5.04.2023

To

the Editors of Tropical Medicine and Infectious Disease®

Dear Editor,

Dear Reviewer,

Thank you for reviewing our manuscript. Please find attached a revised version of our manuscript, “Endocarditis with Streptococcus pseudoporcinus associated with mastocytosis and spondylodiscitis – a coincidental association? A case report?”.

Your and the reviewers’ comments on our revised manuscript were highly insightful and enabled us to greatly improve the quality of our manuscript. We have modified the manuscript in response to the comments. Attached is our point-by-point response to each comment.

Reviewer Comments:

Reviewer 2

The revised version of the manuscript has improved a lot with the addition of new figures, and tables. The authors should work on a few minor things before the final publication.

Answer: Thank you for taking your precious time to be able to reassess our manuscript.

Line 19: Use an abbreviated name for the organism (S. pseudoporcinus) as the full organism name appeared before. Similar in Lines 21, 29, 73, 94, 161, 252.

Answer: Thank you for the suggestion. Done!

Line 38: S. porcinus appears first here. Use the full name.

Answer: We performed the change to full name.

Line 119: Add MRI in the parenthesis after magnetic resonance imaging.

Answer: MRI was added as suggested.

Line 161-163: This is the exact repeat of Lines 94-97.

Answer: Thank you for pointing out this fact. It is our mistake not to delete the sentence after the information from it was used in Table 2.

Line 178: EEG or ECG?

Answer: ECG

ECG image figure number should be changed to Figure no. 3.

According to the organization of the manuscript, ECG figure (Figiure 3) should come first, followed by MRI (Figure 1) and biopsy images (Figure 2).  Figure legends should go after the figures. Italicize the scientific name of the organism in the reference. Figures 1, 2 and Table 2 are not referred to in the texts.

Answer: Thank you for pointing out this fact. We performed the necessary changes as suggested.

We hope that the revised form of the manuscript and our accompanying responses will be sufficient to make our manuscript suitable and accepted for publication in Tropical Medicine and Infectious Disease®. We shall look forward to hearing from you at your earliest convenience.

With our best regards,

Sincerely yours,

Victoria Birlutiu, Prof. Habil. MD. PhD

Rares Mircea Birlutiu, MD PhD

Reviewer 3 Report

Well improved manuscript.

Still persist :

- s instead of sec (table 1) ;

- please define : PVC, PSVC, MV and VTI (figure 2).

Author Response

Sibiu, 10.04.2023

To

the Editors of Microorganisms®

Dear Editor,

Dear Reviewer,

Thank you for reviewing our manuscript. Please find attached a revised version of our manuscript, “Endocarditis with Streptococcus pseudoporcinus associated with mastocytosis and spondylodiscitis – a coincidental association? A case report?”.

Your and the reviewers’ comments were highly insightful and enabled us to greatly improve the quality of our manuscript. We have modified the manuscript in response to the comments. Attached is our point-by-point response to each comment.

Reviewer Comments:

Reviewer 3

Well improved manuscript.

Still persist :

- s instead of sec (table 1) ;

- please define : PVC, PSVC, MV and VTI (figure 2).

Answer: Thank you for taking your precious time to be able to reassess our manuscript. The comments were highly insightful and enabled us to improve our manuscript. Regarding “sec” – we change as requested to “s” a although both abbreviations are used (second (s or sec). PVC and PSVC were defined during the second revision that was submitted prior to receiving your review. MV-VTI is mitral valve velocity time integral and was defined in the text.

We hope that the revised form of the manuscript and our accompanying responses will be sufficient to make our manuscript suitable and accepted for publication in Microorganisms®. We shall look forward to hearing from you at your earliest convenience.

With our best regards,

Sincerely yours,

Victoria Birlutiu, Prof. Habil. MD. PhD

Rares Mircea Birlutiu, MD PhD